



# The cumulative effects of forest disturbance and climate variability on baseflow in a large watershed in British Columbia, Canada

Qiang Li[1], Xiaohua Wei[1*], Mingfang Zhang[2], Wenfei Liu[3], Krysta Giles-Hansen[1], Yi Wang[1],

Liangliang Duan[4]

5    [1]Department of Earth and Environmental Sciences, University of British Columbia Okanagan, 1177
Research Road, Kelowna, British Columbia, Canada, V1V 1V7
[2]School of Resources and Environment, University of Electronic Science and Technology of China,
2006 Xiyuan Ave. Chengdu, China, 611731
[3]Institute of Ecology and Environmental Science, Nanchang Institute of Technology, Nanchang, China
[4]School of Forestry, Northeast Forestry University, Harbin 150040, China

*Correspondence to*: Xiaohua Wei (adam.wei@ubc.ca)

**Abstract.** Assessing how forest disturbance and climate change affect baseflow or groundwater

discharge is critical for understanding water resource supply and protecting aquatic functions. Previous

studies have mainly evaluated the effects of forest disturbance on streamflow, with rare attention on

baseflow, particularly in large watersheds. However, studying this topic is challenging as it requires

explicit inclusion of climate into assessment due to their interactions at any large watersheds. In this

study, we used Upper Similkameen River watershed (USR) (1810 km$^2$), located in the southern interior

of British Columbia, Canada to examine how forest disturbance and climate variability affect baseflow.

The conductivity mass balance method was first used for baseflow separation, and the modified double

mass curves were then employed to quantitatively separate the relative contributions of forest

disturbance and climate variability to annual baseflow. Our results showed that average annual baseflow

and baseflow index (baseflow/streamflow) were about $85.2 \pm 21.5$ mm year$^{-1}$ and $0.22 \pm 0.05$ for the study period of 1954-2013, respectively. The forest disturbance increased the annual baseflow of 18.4 mm, while climate variability decreased 19.4 mm. In addition, forest disturbance also shifted the baseflow regime with increasing of the spring baseflow and decreasing of the summer baseflow. We conclude that forest disturbance significantly altered the baseflow magnitudes and patterns, and its role in annual baseflow was equal to that caused by climate variability in the study watershed despite their opposite changing directions. The implications of our results are discussed in the context of future forest disturbance (or land cover changes) and climate changes.

Key words: Forest disturbance; Climate variability; Baseflow; Relative contributions.

## 1. Introduction

Increasing demands on groundwater resources highlights a critical need for improving knowledge in groundwater recharge and discharge (Scanlon et al., 2006; Uhlenbrook et al., 2002). Baseflow is a critical component in sustaining streamflow particularly during dry periods, and consequently is vital for aquatic habitat and ecosystem functions (Power et al., 1999; Scanlon et al., 2006). Given the importance of baseflow, quantitative assessment of long-term baseflow and its contributing factors is necessary for understanding water balance, groundwater supply and aquatic functions.

In forest-dominant watersheds, the relationships of baseflow with forest and climate changes are usually inferred from the changes in groundwater tables,soil moisture, and streamflow discharges. Several reviews have summarized the effects of forest management on groundwater (Le Maitre et al., 1999; Price, 2011; Smerdon et al., 2009).  The key finding is that after removal of vegetation cover, wetter soil

moisture contents and higher groundwater tables are expected due to loss of evapotranspiration, while gain in vegetation cover results in lower groundwater tables. For example, a ten-year observational study in boreal forests in Eastern Canada showed that the water-table levels after 10-year disturbance were 5-7 cm higher than those of the pre-disturbance levels, but had reached nearly 50% of the pre-cut level (Marcotte et al., 2008).  Clearly, forest changes can greatly affect groundwater recharge and consequently

baseflow. However, most studies regarding the effects of forest disturbance on groundwater recharge or baseflow are conducted for a short period at the small watersheds scale. Direct and quantitative assessment on their long term and cumulative effects in the large watershed scale is rare (Le Maitre et al., 1999; Smerdon et al., 2009).

Climate variability is also another critical factor for influencing baseflow or groundwater recharge variations (Cao and Zheng, 2016; Fleming and Quilty, 2006; Squeo et al., 2006). Forest cover change and climate variability have been recognised as two major drivers influencing hydrological variations in forested watersheds. However, the integrated effects of forest disturbance and climate variability are largely dependent on their individual magnitudes and directions. To understand their individual effects,

various approaches have been used and reviewed (Wei et al., 2013). The paired experimental approach is

commonly used in small watersheds ($<10$ km$^2$) for detecting the effects of forest changes on hydrology as it removes climatic influence, while statistical methods and modelling can be used to separate their relative contributions in large watersheds ($>1000$ km$^2$) simply because the paired experimental approach is not applicable in large watersheds. Although there are growing studies on separating the relative effects of forest change and climatic variability on hydrology, they are mainly on annual streamflow. As far as we know, there are no any studies to quantify the relative contributions of forest disturbance and climate to baseflow in large watersheds.

Direct and quantitative assessment on how baseflow or groundwater discharge responds to both forest and climate changes is a challenging subject. The first challenge is that there are no commonly-accepted methods for baseflow separation even though numerous methods have been developed. As compared with non-trace methods, trace-based methods have been receiving more attentions and applications (Li et al., 2014a; Miller et al., 2014). The tracer-based method uses mass balance to separate streamflow into two end-members: baseflow and surface runoff, based on their different flow paths. Because of longer flow paths, baseflow normally has higher ion concentrations compared with surface runoff (Lott and Stewart, 2016; Matsubayashi et al., 1993; Stewart et al., 2007). Among trace-based methods, conductivity mass balance (CMB) based on close correlations of ion concentration with water specific conductance (conductivity thereafter) is frequently used for baseflow separation. Conductivity data can be readily measured in the field, and is the most effective single parameter for separating surface runoff and ground water contribution to stream (Caissie et al., 1996). As a result, conductivity has been widely used as the tracer for baseflow separation for agricultural watersheds (Li et al., 2014a; Zhang et al., 2013), urban

catchments (Pellerin et al., 2008), natural catchments (Covino and McGlynn, 2007; Miller et al., 2014; Penna et al., 2014), and other environmental settings (Lott and Stewart, 2013; Stewart et al., 2007; Weijs et al., 2013). However, CMB applications are constrained by long-term daily conductivity data, which are not normally available. To address this shortcoming, Miller et al. (2014) recently developed a

regression-based model based on long term discrete conductivity and streamflow data, and then applied this model to estimate long-term continuous daily conductivity using streamflow data.

The second challenge is to separate relative contributions of forest disturbances and climate changes to baseflow in large forested watersheds. Various techniques including sensitivity-based, double mass

curves, simple water balance, and time trend method have been developed for this purpose (Wei et al., 2013). Among them, Wei and Zhang, (2010) adopted modified double mass curves (MDMC) and time series analysis to separate the relative contributions of forest disturbance and climate change to streamflow. As the MDMC is based on water balance in any watersheds, it has the potential to separate the relative effects of forest disturbance and climate variability on baseflow.

The Upper Similkameen River watershed (1810 km$^2$) is a large forest-dominated watershed located in the southern interior of British Columbia, Canada. Over the past a few decades, the watershed experienced various types of forest disturbances (e.g. logging, mountain pine beetle infestation, and wild fire). The cumulative forest disturbance accounts for about 30% of the total watershed area. The river is not

regulated with long-term historic streamflow and climate data. Availability of long-term data, along with

severe forest disturbance provides an excellent opportunity to study the effects of forest disturbance and

climate on baseflow. The key objectives of this study were: 1) to separate and characterize the long-term

baseflow in the study watershed; 2) to quantify the relative contributions of forest disturbance and climate

variability to baseflow; and 3) to discuss possible management implications of our results for water

resource supply and protection of aquatic functions.

## 2. Study site and data

### 2.1. Study site

The Upper Similkameen River at Princeton (USR) is about the 91 km in length and with a drainage area

of 1810 km$^2$. It is located in the southern interior of British Columbia between the Coast Ranges

Mountains and the Okanagan Valley, Canada. The elevation ranges from 630 to 2400 above sea level.

The Similkameen River with its headwater in Manning Park drains to Okanogon River in U.S.A. The

climate across the watershed is characterised by warm summers and cool winters. The watershed covers

several biogeoclimatic zones including Interior Douglas Fir (IDF) zone on the valley floors and Montane

Spruce (MS) and Engelmann Spruce-Subalpine Fir (ESSF) at the higher elevations. The watershed is

underlain by bedrock from several geologic ages. The bedrock types are generally resistant to water

erosion, and form uplands and mountain ranges, which may contain bedrock aquifers, where are highly

fractured.



**Figure 1.** Location, streamflow network, and elevations of the study watershed with the total area of 1810 km$^2$, of which 466 km$^2$ is in USA

## 2.2. Watershed data

### 2.2.1. Climate data

Monthly mean, maximum and minimum temperatures, and precipitation of the study watershed were generated from the ClimateBC dataset (Wang et al., 2006). ClimateBC is a standalone program. It extracts and downscales PRISM (Daly et al., 2008) monthly climate normal data (800 x 800 m) to scale-free point locations, and calculates seasonal and annual climate variables for any specific locations based on latitude, longitude and elevation. Given the large spatial variations in climate, monthly climate data were derived at the resolution of 800 x 800 m, area-weighted climate data were finally derived for the study watershed.

**Figure. 2.** Long-term (1954-2013) average monthly precipitation, and monthly maximum and minimum temperatures in the Upper Similkameen River watershed.

### 2.2.2. Forest disturbance data

Two provincial databases: Cutblocks 2010 and Vegetation Resources Inventory (VRI) 2010 obtained from the British Columbia Ministry of Forests, Lands and Natural Resources Operations, Canada were used to quantify the forest disturbance history in the study watershed. The Cutblocks database records

forest harvesting sizes and timing, while VRI database provides detailed vegetation information including disturbance (i.e. fire, insect infestation, and logging), disturbance timing, biogeoclimatic zones, basal area, etc. Therefore, two databases are complementary and were used for this study.

### 2.2.3. Hydrology and conductivity data

Daily stream discharge data were gained from the hydrometric station (Station ID: 08NL007, Similkameen River at Princeton) operated and maintained by Environment Canada. The annual streamflow were sub-divided into three seasons: spring (March to May), summer (June-October) and fall-winter (November to February). The annual average discharge is 423 mm for the period of 1954 to 2013. The high discharges occurred in spring as a result of snow-melting account for 68% of the annual

discharges.

Specific conductance, streamflow temperature and other water quality data (e.g., copper, zinc, nitrogen and etc.) were collected in the Similkameen River at Princeton on the bridge of Highway 3 near the hydrometric station. Collection of water samples was started from 1966 by Environment Canada (Fig. 3).

There was, however, absence of 10-year (1975-1984, 1986) data collection. For this study a total of 823



samples from 1966 to 2013 were chosen. Specific conductance ranges from 47.4 to 274 μs cm$^{-1}$. Also, in-situ conductivity probe (CTD Diver, DI 271, Schlumberger Water Service, Canada) were installed in the hydrometric station to measure continuous conductivity at the frequency of 30 minutes from May 20, 2015 to September 14, 2016. The conductivity measurements for each day were averaged to derive daily conductivity data.

**Figure 3.** Long term (1966-2013) discrete conductivity measurement and continuous streamflow discharge data in the Upper Similkameen River watershed.

## 3. Methods

### 3.1. Quantifying forest disturbance levels

Logging, Mountain Pine Beetle (MPB) infestation, and wildfire were three major forest disturbance types in the Similkameen River at Princeton (Figs. 4 and 5). According to VRI database, a forest stand in the study watershed was disturbed by either one type (i.e. logging or fire or MPB) or two types of disturbances (logging + Fire or logging+ MPB). Two types of disturbances are defined as a forest stand which is first disturbed by one type followed by the other. A good example is that a forest stand is disturbed by fire first and then by salvage logging.

Different kinds of forest disturbances accumulate over space and time in any forested watersheds. Equivalent clear-cut area (ECA) used to quantify forest disturbance levels is defined as the area that has been clear-cut, fire-killed or infested by MPB, with a reduction factor (ECA coefficient) to account for hydrological recovery due to forest regeneration. An ECA coefficient of 100% means no hydrological recovery in a disturbed area, while an ECA coefficient of 0 indicates a full hydrological recovery. The cumulative clear-cut area (CECA) is the sum of annual ECA values. However, developing ECA data series within a watershed is complicated as hydrological recovery of a forest stand is determined by many factors including disturbance type, climate, and tree species. The detailed estimation of ECA can be found in Wei and Zhang, (2010) and Zhang and Wei, (2012).

Logging or post-disturbance salvage logging is the dominant disturbance type in the study watershed (Fig. 4). The largest logging occurred in 1991 with about 6.4% of the watershed area being harvested. Forest fire happened occasionally. The largest forest fire happened in 1984 with about 1% of the watershed area being disturbed. MPB was not a significant disturbance type until 2003. About 1% and 1.7% of the watershed area were affected by MPB in 2004 and 2007, respectively. Up to year 2011, the cumulative equivalent clear cut area (CECA) was 30% of the total watershed area (Fig. 5). The CECA sharply increased by 8% after the significant logging in 1991. The CECA was then mainly driven by MPB since 2003. In summary, the watershed was heavily disturbed.



**Figure 4.** Annual disturbed area (% of the watershed) in the Upper Similkameen River watershed from 1960 to 2011.

**Figure 5.** Cumulative Equivalent Clear-cut Area (CECA) in the Upper Similkameen River watershed from 1960 to 2011.

## 3.2. Baseflow separation

### 3.2.1. Conductivity mass balance (CMB) method

Daily baseflow data were estimated by the conductivity mass balance (CMB) method, which is expressed as:

$$BF = Q \frac{C_q - C_{ro}}{C_{bf} - C_{ro}} \qquad (1)$$

where, $BF$ is baseflow (m$^3$ s$^{-1}$); $Q$ is daily streamflow discharge (m$^3$ s$^{-1}$); $C_q$ is the conductivity of streamflow (μS cm$^{-1}$); $C_{bf}$ is the conductivity of baseflow (μS cm$^{-1}$); and $C_{ro}$ is the conductivity of surface runoff (μS cm$^{-1}$). $BFI$ is the baseflow index defined as baseflow / streamflow. $C_{ro}$ representing conductivity in streamflow is mainly from surface runoff (i.e. highest flow), while $C_{bf}$ is mainly contributed from baseflow (i.e. lowest flow). Assumptions of applying the CMB method are: 1) contributions from other end-members are negligible; 2) $C_{bf}$ and $C_{ro}$ are constant over the specific period; and 3) $C_{bf}$ and $C_{ro}$ are different from each other (Miller et al., 2014; Stewart et al., 2007).



### 3.2.2. Estimation of daily conductivity

Discrete conductivity data are highly related to daily discharges (Fig. 6). Previous studies indicated that conductivity was also related to time and other variables that describe seasonality and variability in stream

discharges (Miller et al., 2015). A general regression model is showed in Equation (2). This model is characterised with two merits in estimation conductivity. First, it adopts Fourier series to explain seasonality, which reflects the reality and is consistent with snow-melt dominated watersheds. Second, flow anomalies describe flow variability at different time scales. Those two merits add explanatory power and enhance estimation confidence. The model development was constructed in R software (R

Development Team, 2014).

$$
\begin{aligned}
\ln C = I &+ \beta_1 \ln Q + \beta_1 T + \beta_2 \sin(2\pi T) + \beta_3 \cos(2\pi T) \\
&+ \beta_4 \sin(4\pi T) + \beta_5 Cos(4\pi T) + \beta_6 FA
\end{aligned}
\tag{2}
$$

where, $\beta_i$ is coefficient of model parameters; C is daily specific conductance ($\mu S\ cm^{-1}$); I is the intercept of model; Q is daily discharge ($m^3\ s^{-1}$), T is time expressed as decimal years (e.g. 1999.40 = May 26, 1999); and ($\sin(2\pi T) + \cos(2\pi T)$) and ($\sin(2\pi T) + \cos(2\pi T) + \sin(4\pi T) + \cos(4\pi T)$) is simple Fourier

series that explains annual seasonality with one and two concentration annual peaks. FA is flow anomalies (dimensionless). It is calculated from measured daily discharges that describe variability in streamflow at different time scales (e.g. 1-year, 30-day, and 1-day). FA is calculated through R package (Ryberg and Vecchia, 2012).





The best model was evaluated by coefficient of determination ($R^2$), Nash-Sutcliffe Coefficient ($E_{n-s}$), and significance of $P$ values of each modelled variable. The plots of model residuals and fitted values, such as normal probability plots of model residuals, were also evaluated. The model achieving the best performance in the calibration was chosen. The chosen model was then validated with the in-situ

5   continuous conductivity measurement. If the model achieves good estimation with the measured conductivity data, it was then employed as the final model to estimate the continuous conductivity data for baseflow separation (Li et al., 2014b).

**Figure 6.** The relationship between streamflow conductivity (Y) and streamflow discharge (X) in the Upper Similkameen River watershed.

### 3.2.3.   Selection of $C_{bf}$ and $C_{ro}$

The accuracy of baseflow estimation in the CMB method is highly dependent on the selection of two parameters ($C_{bf}$ and $C_{ro}$). Like other baseflow separation methods, all contributed end-members are lumped into two broadly components (i.e. baseflow and surface runoff). Separation of a hydrograph into

15   two end-members is more appropriate for large watersheds (>1000 km$^2$), as compared with smaller watersheds where more than two-members should be considered (Uhlenbrook et al., 2002). In addition, Zhang et al. (2013) reported that baseflow is more sensitive to $C_{bf}$ rather than $C_{ro}$ in the small snow-dominated watershed (14.5 km$^2$). Also, in-situ conductivity measurements in 14 large watersheds (>1000 km$^2$) indicated that the constant $C_{ro}$ might not have large impacts on the baseflow (Miller et al., 2015).

20   The annual paired $C_{bf}$ and $C_{ro}$ were selected for each water year (i.e. August to September) to minimize

temporal variations in conductivity. Conductivities corresponding to 0.01 and 0.99 quantile flow rates (i.e. flow exceeds at 1% and 99% of the time in a given year) in one year were averaged and treated as annual $C_{bf}$ and $C_{ro}$, respectively.

### 3.3. Cross correlations analysis

Cross correlations analysis is an effective and widely used method to test if there are significant relationships among time series variables. Its advantage is that it can remove autocorrelations existing in data series and identify lagged causality between two data series (Zhang and Wei, 2014). In this study, it was adopted to detect the relationships and lagged effects between forest disturbance and hydrological variables. All the tested hydrological variables and forest disturbance data were pre-whitened to remove the autocorrelation by fitting the ARIMA (Autoregressive Integrated Moving Average) models. Model residuals from ARIMA models with best performance, achievements of model stationary and coefficient of determination were selected for cross-correlation tests (Liu et al., 2015a).

### 3.4. Quantification of relative contributions of forest disturbance and climatic variability to baseflow

Forest disturbance and climate variability are commonly considered as the two major drivers for hydrological variations in large forested watersheds. The modified double mass curves (MDMC) were normally used to quantify the relative contributions of forest disturbance and climatic variability to annual streamflow (Liu et al., 2015a; Zhang and Wei, 2012; Zhang et al., 2012). To quantify the relative



contributions of forest disturbance and climatic variability to baseflow in this study, we applied the same approaches as we did for annual streamflow. The details for assessing the relative contributions of forest disturbance and climatic variability to annual streamflow are described in (Liu et al., 2015a; Wei and Zhang, 2010; Zhang and Wei, 2012; Zhang et al., 2012).

Streamflow is divided into surface runoff (*RO*) and baseflow (*BF*). Theoretically, water balance in large watersheds can be expressed as: *P = ET + RO + BF* (3a) or *BF = P- ET- RO* (3b). The effective groundwater discharge ($G_e$) is then defined as the difference between precipitation, evapotranspiration, and surface runoff. A liner relationship is assumed between accumulative baseflow ($BF_a$) and

10 accumulative effective groundwater discharge ($G_e$). Thus, MDMC can be modified by plotting $BF_a$ against cumulative $G_e$. Breaking points can be identified on the MDMC if there are significant influences from non-climatic variables. Thus, the relative effects of forest disturbance and climatic variability on baseflow can be quantified. Relative contributions of forest disturbance and climate change to baseflow can be calibrated as:

$$R_f = \frac{\left|\Delta Q_f\right|}{\left|\Delta Q_f\right| + \left|\Delta Q_c\right|} \times 100\% \tag{4}$$

$$R_c = \frac{\left|\Delta Q_c\right|}{\left|\Delta Q_f\right| + \left|\Delta Q_c\right|} \times 100\% \tag{5}$$





where, $R_f$ and $R_c$ are the relative contributions of forest disturbance and climate variability to baseflow, respectively. $\Delta Q_f$ and $\Delta Q_c$ are the deviations of annual baseflow caused by only forest disturbance and climate change, respectively.

5   In this study, monthly potential evapotranspiration (*PET*) was estimated through Eqn. (6) and then used to calculate actual evapotranspiration (*ET*) by both Eqns. (7) and (8). The final monthly *ET* estimates were averaged from those two methods or equations.

$$PET = 0.0023\, R_a \left[ \frac{T_{\max} + T_{\min}}{2} + 17.8 \right] (T_{\max} - T_{\min})^{0.5} \tag{6}$$

$$ET = \{ P[1 - \exp(-PET/P] \times PET \times \tanh(P/PET) \}^{0.5} \tag{7}$$

$$ET = P \frac{1 + \omega(PET/P)}{1 + \omega(PET/P) + P/PET} \tag{8}$$

where, Eqns. (6), (7), and (8) are Hargreaves (Hargreaves and Allen, 2003), Budydo (Budyko, 1974) and Zhang (Zhang et al., 2001), respectively. $R_a$ is extraterrestrial radiation (MJ m$^{-2}$); *PET* is potential evapotranspiration (mm); $T_{max}$ and $T_{min}$ are maximum and minimum temperature (°C); $P$ is precipitation (mm); *ET* is actual evapotranspiration (mm); $w$ plant-available water coefficient ($w$ =2 used for this study).

15   **4.   Results**

**4.1. Baseflow from regression-based conductivity estimation**

The parameters of the conductivity estimation model were finalized as: logarithm of streamflow discharge (Ln Q), timing (T), two concentration peaks of Fourier series (sin (2πT) + cos (2πT) + sin (4πT) + cos



($4\pi T$)), and flow anomalies (FA) with time scales of 1 year, 30 days and 1 day. The coefficients of the regression model variables were listed in Table 1. The diagnostic statistics including $R^2$ and $E_{n-s}$ between the simulated and observed conductivity values are 0.87 and 0.86, respectively. The established regression model was then used to calculate conductivity (sample size: 118) from May 20, 2015 to

September 14, 2015 for validating the model performance. The $R^2$ is 0.91, while the root mean squared error is 17.8 µs cm$^{-1}$ and $E_{n-s}$ is 0.91. The results suggest that the regression model can be reasonably well used to estimate continuous conductivity.

Using the established model, we estimated that average annual baseflow rate and BFI (baseflow index)

were $85.2 \pm 21.5$ mm year$^{-1}$ and $0.22 \pm 0.05$ for the period of 1954-2013, respectively (Fig. 7). The lowest and highest baseflow rates were 41.8 mm year$^{-1}$ in 1993 and 127.1 mm year$^{-1}$ in 1972, respectively, while the lowest and highest BFI were 0.14 in 1971 and 0.35 in 2001, respectively. The higher monthly baseflow rates were found in the snow-melting seasons (e.g., April (10.8 mm month$^{-1}$) and May (13.5 mm month$^{-1}$)) (Fig. 8).

**Table 1.** Summary of the regression model for conductivity estimation

**Figure 7.** Long-term annual mean streamflow, baseflow, and baseflow index (BFI) in the Upper Similkameen River watershed from 1954 to 2013.





**Figure 8.** Long-term monthly streamflow, baseflow, and baseflow index (BFI) in the Upper Similkameen River watershed from 1954 to 2013.

## 4.2. Cross-correlations between CECA and hydrological variables

The cross-correlations between the CECA and hydrological variables in Table 2 showed that forest disturbance had significantly affected hydrological variables. Forest disturbance significantly increased the annual and spring baseflows, but it significantly decreased the summer baseflow. No significant impacts of forest disturbance on winter baseflow were found in the study watershed.

**Table 2.** Cross-correlations between cumulative equivalent clear-cut area (CECA) and hydrological variables

## 4.3. Quantification of the relative effects of forest disturbance and climate on annual baseflow

Only one breaking point (year 1972) was detected in the MDMC for baseflow (Fig. 9). The fitted ARIMA intervention test for the slope of MDMC also showed the significant intervention occurred in 1972 (Table 3). The whole study period was, therefore, divided into the reference period (1954-1972) and the disturbed period (1973-2013). As shown in Fig. 9, in the reference period, a straight line indicated that cumulative annual baseflow changes were consistent with cumulative groundwater discharge variations. For the

disturbed period, the distinct shift in 1972 was found between the observed and predicted cumulative

baseflow, indicating the significant effects of forest disturbance on baseflow. For a better explanation of

temporal variations of those effects, the disturbed period were subsequently sub-divided into five periods

listed in Table 4.

Fig. 10 also shows large annual baseflow deviations ranging from -127.5 to 306.9 mm across those five

disturbed periods. In addition, $R_f$ to baseflow steadily increased with the CECA increase. With the forest

disturbance level of 2.31% in the period of 1973 to 1982, climate variability was the dominate factor for

the annual baseflow variations, and its relative contribution accounts for 62.5% (or 13.3 mm) of the total

10   annual baseflow variations.  The effects of forest disturbance on baseflow then became dominate with the

CECA increase in the period of 1993 to 2013. Overall, forest disturbance increased annual baseflow of

18.4 mm, while climate variability decreased 19.4 mm in the disturbance period (1973-2013).  Our result

also clearly demonstrate that forest disturbance and climate variability played a co-equal role for baseflow

with $R_f$ and $R_c$ being 48.8% and 51.2% of the total baseflow variations, respectively.

**Figure 9.**  Modified Double Mass Curve of cumulative annual baseflow vs. cumulative groundwater
discharge

**Table 3.** ARIMA Intervention for the slope of MDMC for baseflow

**Figure 10**. Annual variations of the effects of forest disturbance on annual baseflow in the Upper Similkameen River watershed from 1973 to 2013.

**Table 4.** Relative contributions of forest disturbance and climate variability to annual baseflow in the Upper Similkameen River watershed from 1973 to 2013.

## 5. Discussions

### 5.1. Baseflow estimation

Our baseflow separation results from the tracer-based method show that average annual baseflow rate and BFI were $85.2 \pm 21.5$ mm year$^{-1}$ and $0.22 \pm 0.05$ for the period of 1954-2013, respectively. Those estimates are comparable with groundwater recharges estimated by other studies in the study region. For examples, annual recharge rate in Grand Folks is about 135.46 mm year$^{-1}$ (precipitation: 471 mm) calculated through the groundwater model validation (Allen et al., 2004). Average annual recharge rates are 88 mm (precipitation: 447 mm) in Vernon (Liggett and Allen, 2010) and $77.8 \pm 50.8$ mm (precipitation at $496.5 \pm 77.9$ mm) in Deep Creek watershed, northern Okanagan (Assefa and Woodbury, 2013). Given the spatial variations in climate, land use and geology, our estimations on baseflow are acceptable in our study watershed.



## 5.2. The cumulative effects of forest disturbance on annual baseflow

Over the disturbed period of 1973-2013, the average increment of annual baseflow attributed to forest disturbance was 18.4 mm, which accounted for 30.8% of total annual baseflow. The baseflow changes were low for the period of 1973 to 1982 when the forest disturbance level was small (2.31%). The

5   baseflow increased dramatically after 1991 with the significant logging happened in 1991 and the mountain pine beetle infestation broke out in 2003. Clearly, the cumulative forest disturbance in the study watershed significantly increased annual baseflow. Such increasing is likely due to increasing of soil water storages and consequent groundwater recharging as a result of less evapotranspiration caused by forest disturbance.

Our results on the effects of forest disturbance on annual baseflow are consistent with other studies. For examples, after the native forest removal, the groundwater recharge increased from 1.5 to 6 mm year$^{-1}$ in a large watershed (1250 km$^2$) located in the west Australia (Leaney and Herceg, 1995). Clearing the native vegetation in the Murry River basin has increased the recharge from <0.1 mm to 0.2 mm year$^{-1}$ to 3 to 30

15   mm year$^{-1}$ (Barnett, 1990). In the central interior of British Columbia, Canada, Rex and Dubé, (2006) concluded that the groundwater level has been elevated by 10 cm following by forest harvest than mountain pine beetle infestation. In Vancouver Island, Canada, the groundwater table has been increased 30-50 cm after 10 years of logging (Hetherington, 1998). Despite those consistent results in terms of changing directions, there are large variations in changing magnitudes. This is because of the differences

20   in climate, spatial heterogeneity of geologic and soil conditions, groundwater storage, and forest disturbance levels among those studied watersheds. Therefore, that the effects of forest disturbance on





baseflow are likely watershed specific and more case studies are needed to develop general conclusions

regarding the relationship between forest changes and baseflow.

As mentioned in the introduction, previous studies on the similar topic often took an indirect approach to

assess the effects of forest change on baseflow, and consequently there is a general lack of quantitative

and direct assessment on the cumulative effects of forest disturbance on baseflow. In this study, our

MDMC framework was successfully applied to quantify the long term cumulative effects of forest

disturbance on baseflow, which can be extended to other watershed studies where similar data are

available. The MDMC framework has been successfully employed to separate the effects of climate and

vegetation changes on annual streamflow in different climatic regions (Liu et al., 2015a; Yao et al., 2012;

Zhang et al., 2012). It was our first application of this framework on separating the relative effects of

forest disturbance and climate change on annual baseflow. To implement this methodology for assessing

forest disturbance and baseflow, we need to generate a relative accurate baseflow data series. In this study,

we used the objective tracer-based method for baseflow separation, which minimized the uncertainties in

baseflow separation methods. In addition, long term data on climate, hydrology and forest change must

be available in any study watersheds where forests experience significant changes (e.g., disturbance or

reforestation) so that their significant hydrological effects can be detected.

## 5.3. The cumulative effects of forest disturbance on seasonal baseflow

Cross-correlation results revealed that forest disturbance has altered the baseflow regimes, specifically, increased spring baseflow while decreased summer baseflow in our watershed (Table 3). Similar results have been found in other studies (e.g. Robinson and Dupeyrat, 2005; Eisenbies et al., 2007). This is likely

due to higher rates of snow accumulation and melting following forest disturbance (Winkler et al., 2015). Such increases may add additional water sources for spring baseflow but reduce water recharging for summer baseflow because forest disturbance may advance the snow melting timing. Thus, the baseflow in summer was relatively reduced as compared to those in the pre-disturbance periods. The similar results were also reported by the paired experimental watersheds (Winkler et al., 2015) and the hydrological

modelling study (Schnorbus and Alila, 2013) in the same region.

Low flow (baseflow) in dry seasons has been receiving growing attentions due to increased demands. In snow-dominated watersheds, forest disturbance increased low flow in Baker River watershed (1560 km$^2$), while no significant changes on low flow were found in Willow River watershed (3185 km$^2$) with a similar

level of forest disturbance (Zhang et al., 2015). In rain-dominated watersheds, deforestation reduced the low flows by 30% in Meijiang watershed (6983 km$^2$) (Liu et al., 2015b). Bruijnzeel, (2004) reviewed several studies in tropical regions indicating that forest harvesting decreased the low flow in dry seasons. Zhou et al. (2010) found that the low flow in dry seasons has been significantly increased by reforestation in Guangdong Province (179752 km$^2$), subtropical of China. The above inconsistent results indicate that

low flow responses to forest change are highly variable as they are not only related to the level of forest changes but also the alteration of soil conditions.

### 5.4. Offsetting effects of forest disturbance and climate change on baseflow

Various studies have showed that forest disturbance and climate change can play offsetting effects on annual streamflow (Wei and Zhang, 2010; Zhang et al., 2012). Our study also demonstrated that forest

disturbance and climate variability played similar roles on annual baseflow, but with opposite directions: forest disturbance increased annual baseflow, while climate variability decreased it in the study period. However, there are temporal variations in relative contributions of forest disturbance to annual baseflow depending on the levels of forest disturbance. In our study, we found that the impacts of forest disturbance on baseflow are larger than those from climate change in the periods of 1993-2013, while smaller in the

periods of 1973-1992. The similar results were also reported by other studies. For example, the study conducted in the Nebraska Sand Hills, USA found that land use change is more significant than climate change on baseflow (Wang and Cai, 2010). A comprehensive study by Juckem et al. (2008) found that climate change has advanced baseflow timing, while land cover change significantly altered the baseflow quantity.

The effects of forest disturbance and climatic variability on annual baseflow can also be additive. While offsetting effects lead to less variation in water resource, additive effects can cause river flows to either increase (e.g., higher chances of floods) or decease (e.g., higher chances of droughts). Thus, when managing any watersheds for future water resources, forest changes, climate variability and their

interactions should be carefully considered.

## 5.5. Implications of forest disturbance on watershed management

Groundwater is important for maintaining aquatic ecosystems and providing critical water supply in dry periods (Power et al., 1999). Although forest disturbance increased annual mean baseflow, it significantly reduced summer baseflow when water demands are the most in the study region. The reduced summer baseflow, along with higher temperature in the summer can intensify the pressures on aquatic systems, which may negatively affect the habitat and life stages of various salmon fish species in the study watershed (Hyatt et al., 2003). In addition, the reduced summer baseflow will also intensify competitions on water resources between human demand and those for maintaining aquatic functions. On the other side, with consideration of the water scarcity in the region, the forest disturbance increased the annual baseflow, which can attenuate the water stress in the region caused by climate change. Thus, forest disturbance, climate and their interactions must be carefully managed in order to sustain water supply for human being as well as for aquatic functions.

## 6. Conclusions

Quantifying the long term and cumulative effects of forest disturbance on baseflow has rarely been reported in the literature. From this study, we conclude that forest disturbance significantly increased annual baseflow, while climate variability decreased it. In addition, forest disturbance also altered seasonal baseflow patterns by increasing the spring baseflow and decreasing the summer baseflow. All

those hydrological effects on baseflow have important implications for protecting water supply and aquatic systems, which should be carefully managed.

## Acknowledgements

We thank the British Columbia Ministry of Forests, Lands and Natural Resources Operations for providing forest inventory data. Thanks are also given to Environment Canada for their streamflow and conductivity data. The funding for supporting this project was provided by the Regional District of Okanagan-Similkameen through a contract.

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





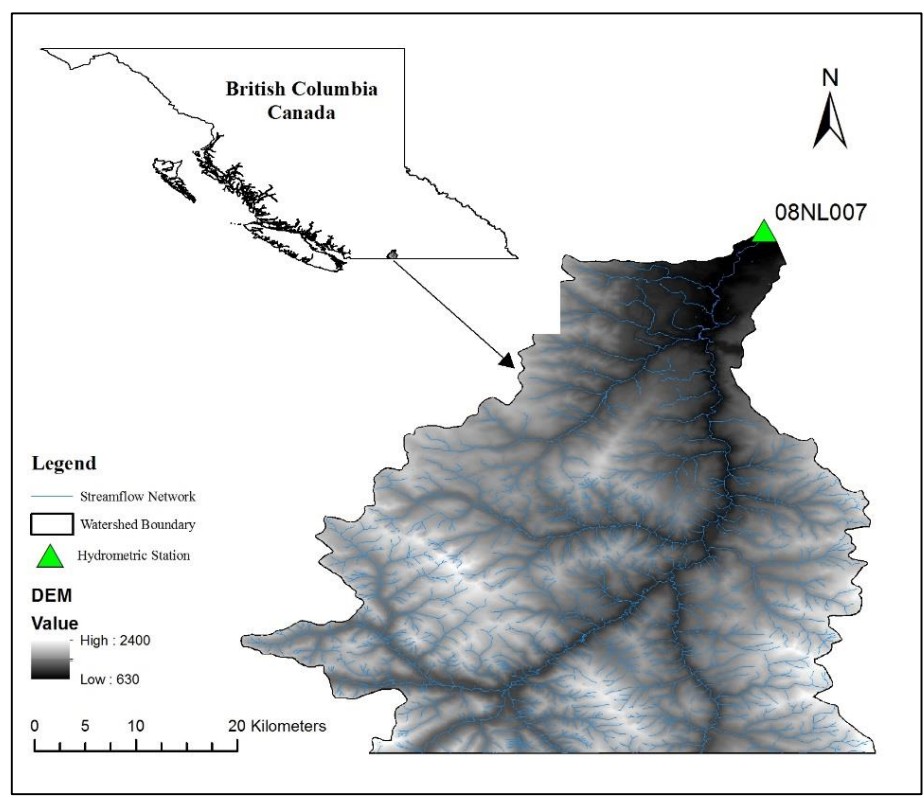

**Figure 1.** Location, streamflow network, and elevations of the study watershed with the total area of 1810 km$^2$, of which 466 km$^2$ is in USA





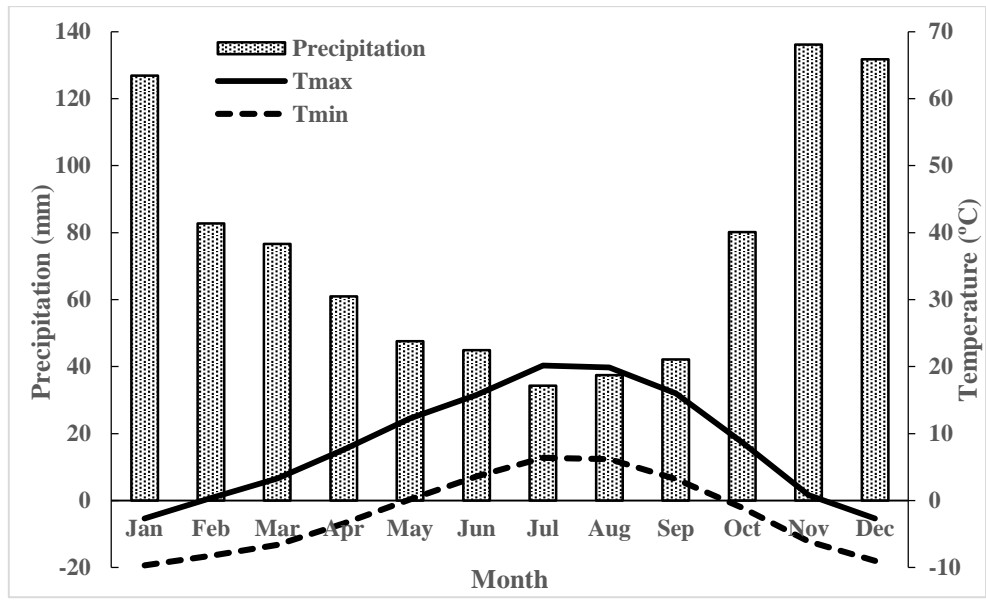

**Figure 2.** Long-term (1954-2013) average monthly precipitation, and monthly maximum and minimum

temperatures in the Upper Similkameen River watershed





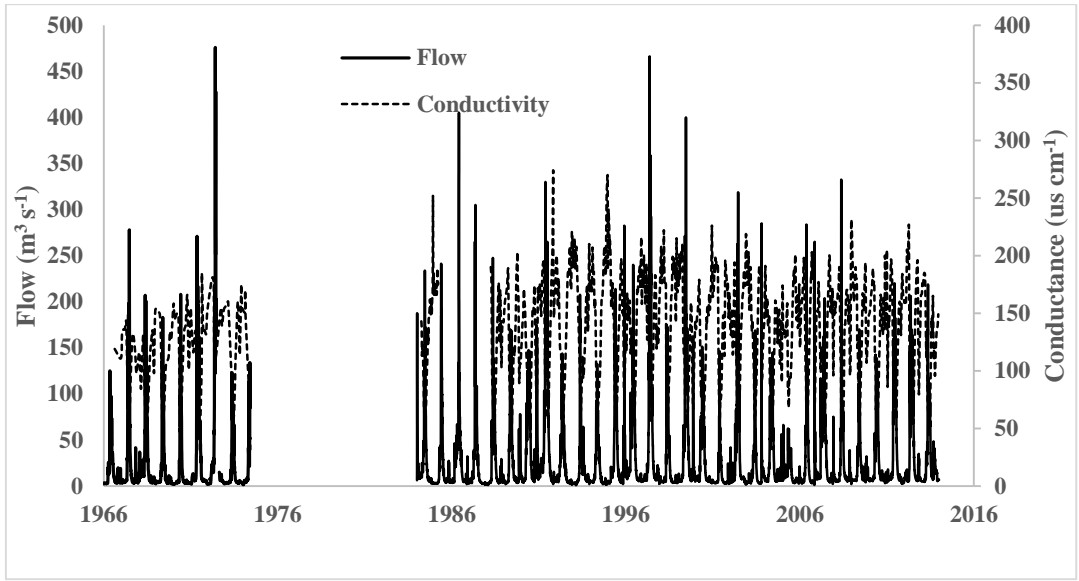

**Figure 3.** Long term (1966-2013) discrete conductivity measurement and continuous streamflow discharge data in Upper Similkameen River watershed.





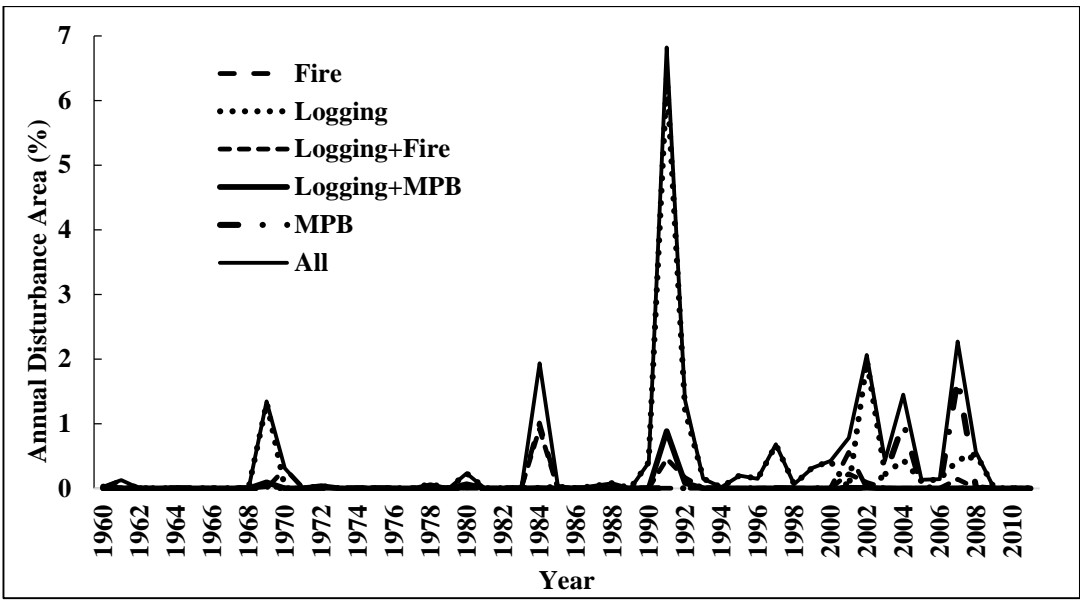

**Figure 4.** Annual disturbed area (% of the watershed) in the Upper Similkameen River watershed from 1960 to 2011.





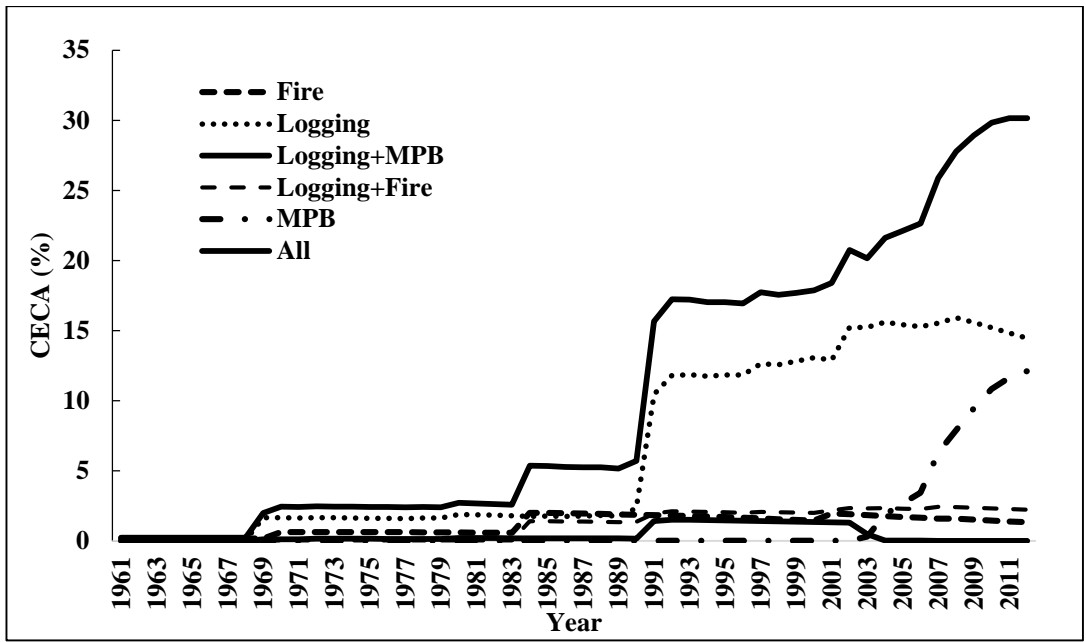

**Figure 5**. Cumulative Equivalent Clear-cut Area (CECA) in the Upper Similkameen River watershed from 1960 to 2011.





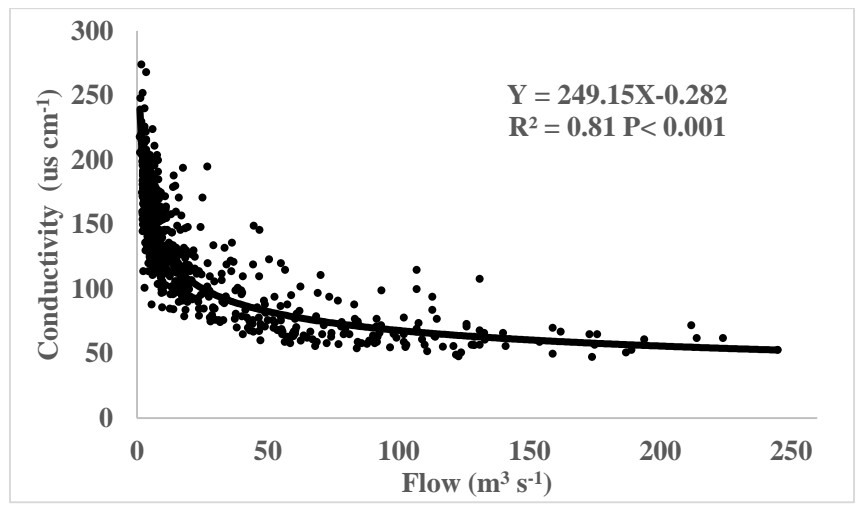

**Figure 6.** The relationship between streamflow conductivity (Y) and streamflow discharge (X) in Similkameen River at Princeton.

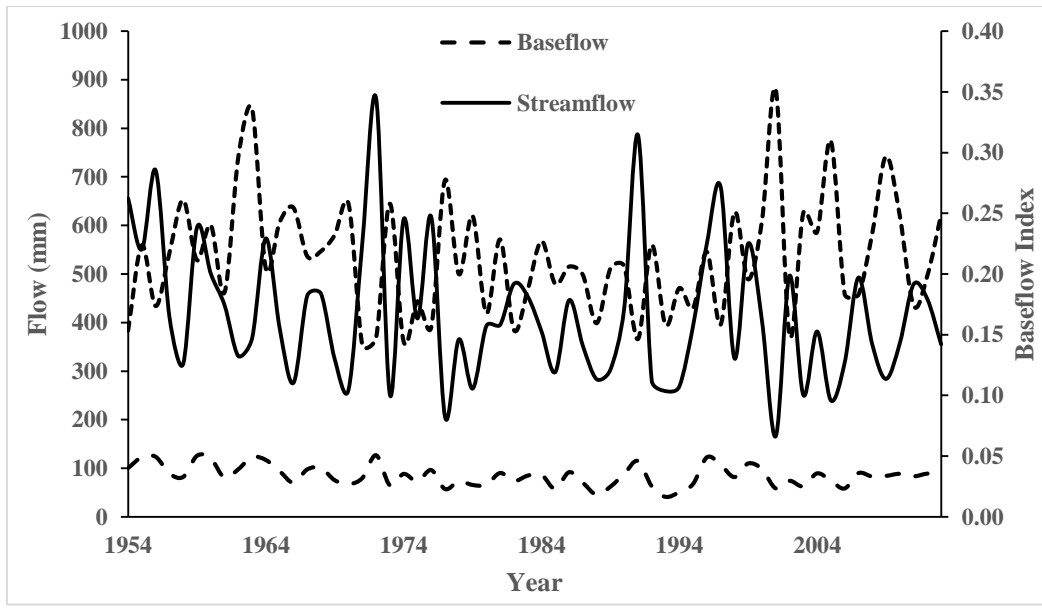

**Figure 7.** Long-term annual mean streamflow, baseflow, and baseflow index (BFI) in the Upper Similkameen River watershed at Princeton from 1954 to 2013.





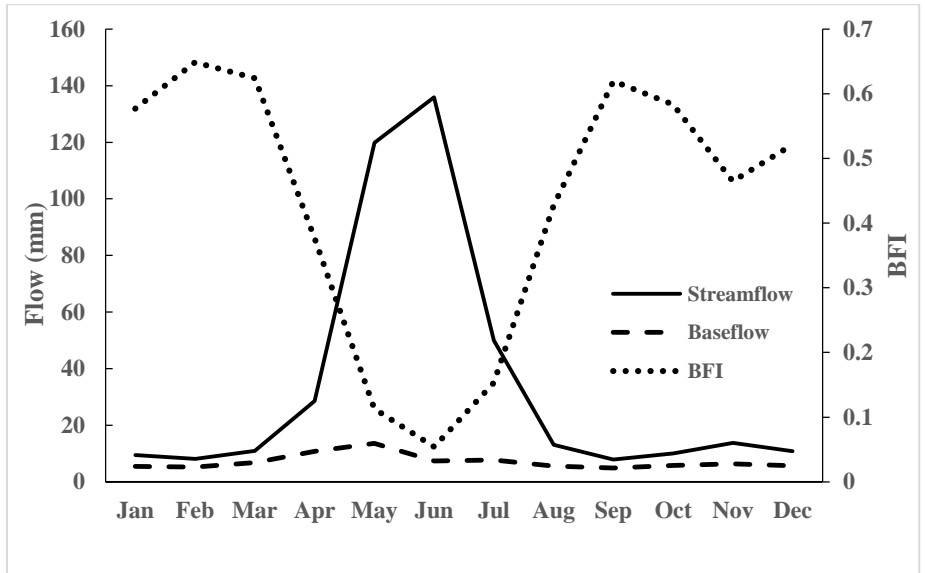

**Figure 8.** Long-term monthly streamflow, baseflow, and baseflow index (BFI) in the Upper Similkameen River watershed at Princeton from 1954 to 2013.

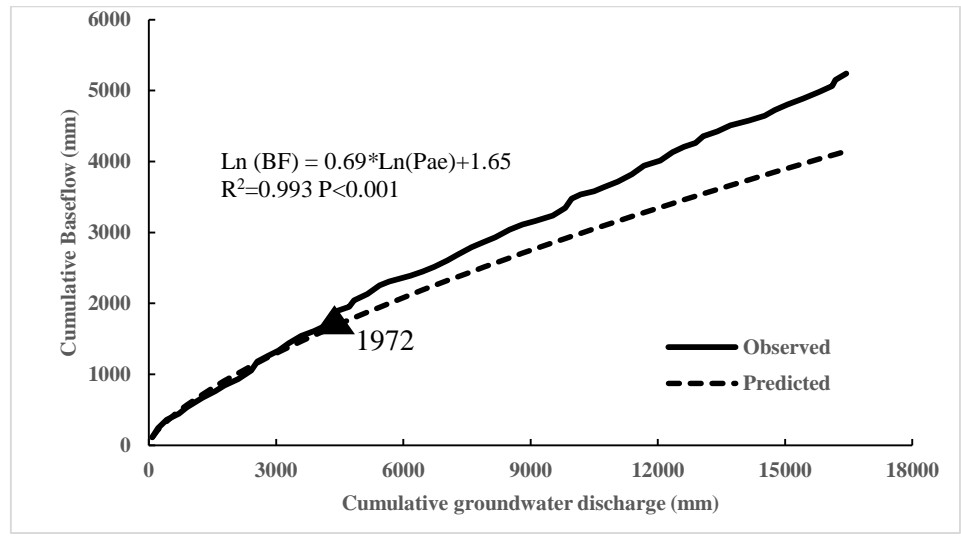

**Figure 9.** Modified Double Mass Curve of cumulative annual baseflow vs. cumulative groundwater discharge





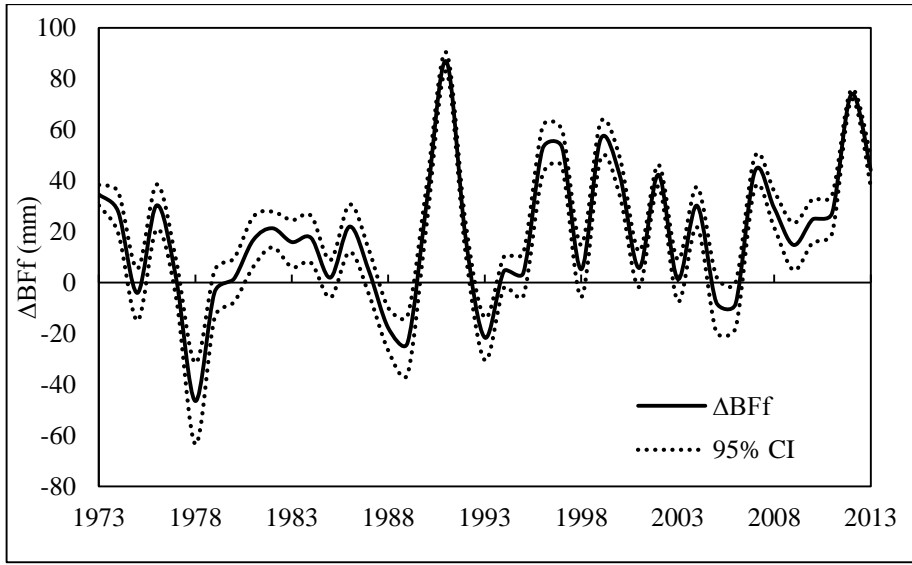

**Figure 10**. Annual variations of the effects of forest disturbance on annual baseflow in the Upper Similkameen River watershed from 1973 to 2013.



**Table 1.** Summary of the regression model for conductivity estimation

| Parameters | Estimate | Std. Error | T value | P |
|---|---|---|---|---|
| Intercept | -2.334 | 0.867 | -2.693 | <0.01 |
| Ln Q | -0.243 | 0.012 | -19.684 | <0.001 |
| T | 0.004 | 0.000 | 8.865 | <0.001 |
| $\sin(2\pi T)$ | 0.054 | 0.007 | 7.552 | <0.001 |
| $\cos(2\pi T)$ | 0.004 | 0.009 | 0.434 | 0.665 |
| $\sin(4\pi T)$ | -0.027 | 0.007 | -3.801 | <0.001 |
| $\cos(4\pi T)$ | -0.026 | 0.007 | -3.527 | <0.001 |
| FA_1 year | 0.068 | 0.019 | 3.648 | <0.001 |
| FA_30 days | -0.132 | 0.026 | -4.994 | <0.001 |
| FA_1 day | 0.037 | 0.032 | 1.153 | 0.249 |





**Table 2.** Cross-correlations between cumulative equivalent clear-cut area (CECA) and hydrological variables

| Hydrological Variables | Cross-correlation | | |
|---|---|---|---|
| | ARIMA Model | Cross-correlation coefficients | Lags |
| Annual mean flow | Ln, (0, 1, 1), Lag1 | **0.34\*** | -7 |
| Annual mean baseflow | Ln, (0, 1, 1), Lag1 | **0.43\*** | -7 |
| Spring mean baseflow | Ln, (1, 1, 1), Lag1 | **0.37\*** | -7 |
| Summer mean baseflow | Ln, (2, 1, 1), Lag1 | **-0.39\*** | -1 |
| | | **-0.38\*** | -10 |
| Fall-Winter mean baseflow | Ln, (2, 1, 1), Lag1 | -0.19 | 0 |
| ARIMA model for CECA, model structure: Ln, (0, 2, 1) Lag 1; | | | |



**Table 3.** ARIMA Intervention for the slope of MDMC for baseflow

| Model Input | Model Structure | Parameters Estimation | | |
|---|---|---|---|---|
| | | Intervention Type | Q (1) | Ω (1) |
| Slope of MDMC for baseflow | Interrupted ARIMA: (0, 3, 1), Intervention at year 1972 | Abrupt Permanent | 0.96 (P<0.001) | 0.011 (P= 0.02) |

**Table 4.** Relative contributions of forest disturbance and climate variability to annual baseflow in the

5  Upper Similkameen River watershed from 1973 to 2013.

| Baseflow | ΔBF (mm) | $\Delta BF_f$ (mm) | $\Delta BF_f/BF$ (%) | $\Delta BF_c$ (mm) | $\Delta BF_c/BF$ (%) | $R_f$ (%) | $R_c$ (%) | CECA (%) |
|---|---|---|---|---|---|---|---|---|
| 1973-1982 | -5.3 | 8 | 12.0 | -13.3 | -19.9 | 37.5 | 62.5 | 2.31 |
| 1983-1992 | -1.1 | 15.8 | 26.1 | -16.9 | -27.9 | 48.3 | 51.7 | 5.48 |
| 1993-2002 | 1.1 | 24.4 | 43.0 | -23.3 | -41.1 | 51.2 | 48.8 | 14.33 |
| 2003-2013 | 1.3 | 24.9 | 41.1 | -23.6 | -38.9 | 51.3 | 48.7 | 23.94 |
| 1973-2013 | -0.9 | 18.4 | 30.8 | -19.4 | -32.4 | 48.8 | 51.2 | 11.82 |