# Peer review of "The cumulative effects of forest disturbance and climate variability on baseflow in a large"

_Hydrology and Earth System Sciences, 2016_

## Referee Comment (RC1) · Anonymous Referee #1 · 8 Jul 2016

General comments:

The main aim of the paper is to reveal or distinguish the individual contributions of forest disturbance and climate variability on baseflow in a forest catchment. However, the manuscript was poorly organized that so many uncertainties are in their methods and discussions.

The authors adopted several distinct methods in their study just like the baseflow separation method which seems to be critical in their study, and the MDMC method for quantifying the relative contributions of forest disturbance and climate variability and even Budyko method has also been adopted in this study. However, the discussions for each method were not enough, which lead to many uncertainties and questions to

the readers. For example, the baseflow separation method, CMB developed based on Pinder and Jones [1969], was firstly proposed to investigate the variation in the runoff composition in three small drainage basins which are relatively simplex in flow path and components compared with larger catchments. As to the MDMC method, it needs a logistic flow chart to describe the whole processes, and more details should be provided on how you calculate $\Delta Qf$ and $\Delta Qc$.

Another important issue is that how climate has changed in this catchment based on the observations and has climate changes impacted the forest itself about the coverage rate, beetle infestation?

The figures and tables are in poor quality.

Based on above considerations, especially the poor organization and insufficiency of discussions on various adopted methods, I have to recommend "major revisions", would like to encourage resubmission.

Specific Comments:

P means page, and L means lines

P1L14: what is the different between baseflow and groundwater discharge.

P1L17: how do you define a catchment a large one?

P1L17-23: As you have mentioned "However, studying this topic is challenging as it requires explicit inclusion of climate into assessment due to their interactions at any large watersheds." How do you think that CMB method can solve this problem?

P2L11: 'Introduction' is poorly written for that the method seems not to satisfy your main work and there so many uncertainties in your study. For instance, larger catchments tend to be disturbed by human activities in their wide downstream alluvial plain, and the hydrological responses seem to be lagged by drainage network and inapparent compared with small catchments. As we know, CMB was developed based on

scientific studies on smaller catchment, how do your upscale the method for applying in large scale catchment?

P3L8: "......but had reached nearly 50% of the pre-cut level......" of WHAT?

P4L4-7: "Although there are ...... annual streamflow. ......in large watersheds." What do you mean here? What is the logistic relation? I have expected as "There are many studies on annual streamflow but no study on monthly or daily streamflow". Anyway, critical comments in Introduction should be better if supported by data and comments of other colleagues.

P4L10: "there are no commonly-accepted " As there are no commonly-accepted methods for baseflow separation, how do you verify that your adopted method is suitable for the study catchment.

P6L12: "......Manning Park drains to Okanogon River in U.S.A." I found that the river drains to Canada in figure 1?

P6L16-18: "The bedrock types are generally resistant to water erosion, and form uplands and mountain ranges, which may contain bedrock aquifers, where are highly fractured." It seems to be inconsistent!! As it hard to be eroded, why is fractured and contains aquifers?

P8L1: how long have the forest disturbance databases been recorded?

P8L20: "......a total of 823......" Is the water quality data measured per month?

P9L11: how about the distribution of forest disturbance? Is it changed every year? How to evaluation the change of position of forest disturbance on baseflow?

P11L8: CMB was developed based on scientific studies on smaller catchments. While applying in large catchments, other factors and processes, e.g., effects of exchange between riparian and river, storage and lag in river network, shall be considered. So CMB method in equation (1) may have to be improved before applying in large catchments.

P12L12-18: all the variables should be in italic.

P13L12-16: this paragraph is difficult to be understood and need to be re-written.

P13L19: As you only use one equation (Equation (1)) for baseflow separations, I could not figure out why will Cro play so different effects on baseflow in small and large catchments? As the smaller or larger value of Cro will impact the value of BF no matter what values of Q is adopted in terms of the scale of catchment areas. It is worth to note that Cro determines the ratio of BF ad Q.

P14L15: Add a flow chart about MDMC.

P15L6-9: shall equation (3) be placed in individual line? How do you calculate Ge and BFa in equation (3)?

P16L5-14: what is the relationship between PET and MDMC?

P16L11: Budydo should be Budyko. w is $\omega$?

P16L16: how do you determine the values of Cro and Cbf in Section 4.1? As you defined them as constant, thus how do you derive their values?

P18L14: in figure 9, I really can hardly figure out the breaking point at 1972. The authors need to display more data in figures or tables. What is the difference between the baseflow and groundwater discharge in the vertical and horizontal axis in figure 9?

P23L2-3: in table 3, cross-correlation results revealed that forest disturbance has altered the baseflow regimes, specifically, increased spring baseflow while decreased summer baseflow in our watershed? I also cannot find the trends.

P24L1: provide a figure to show how the climate changes or fluctuates.

P25L15: conclusions is too short and a deepen summary and perspective are needed.

The references should be carefully revised according to requirements of HESS. Some

of the references are lack of Doi (e.g., Weijs et al., 2013 and Winkler et al., 2014) and others may use wrong symbols (e.g., Power et al., 1999).

Figure 1 should be revised by adding a complete map about the catchment though it belongs to two countries. Land cover map also needs to be added to show the forest distributions.

Figure 5, in this figure, I find that CECA in some curves have little decreases as a function of year. Please try to explain why this happens.

Figure 7, it is lack of BFI in the legend.

Figure 9, what is the difference between baseflow and groundwater discharge?

Figure 10, how do you compute $\Delta$BFf ? why not add $\Delta$BFc in the same figure? Is $\Delta$BF a bias between 1973 to 2013 and the reference period (before 1972)? There are not logging or beetle infestations in the reference period, are there? If no, how do you define it as the reference period?

References: Pinder and Jones [1969], Determination of the Ground-Water Component of Peak Discharge from the Chemistry of Total Runoff, Water Resources Research, 5(2), 438-445.

---

## Referee Comment (RC2) · Anonymous Referee #2 · 18 Jul 2016

1. General comment: In this paper, the authors applied the Budyko based method to analyze the effect of climatic variability and forest disturbance on the annual runoff, and then separate the change of the baseflow from the total change of the annual runoff by using conductivity mass balance method. In general, it's hard to say that this paper has any strong innovation point, since it is just a case study based on two common methods. The authors just consider the influence of climatic variability and forest disturbance on the rainfall-runoff relationship, but ignore the possible variation in the relationship between baseflow and total runoff caused by the forest disturbance. The results of paper are questionable. Leaving aside the lack in the methodological contribution, this paper also has a very bad presentation. I think this paper cannot be

accepted by this journal.

2. Specific comments: (1) Line 31: "85.2±21.5", "0.22±0.05". I wonder how the terms 21.5 and 0.05 were calculated. The authors presented neither the calculation method nor the specific meaning of these terms in the paper. The similar issues in other parts of this paper are also obvious. (2) Line 35: What is the specific meaning of forest disturbance? Increase or decrease in forest cover? (3) Section 2: The study watershed in this paper spans USA and Canada, but the authors just consider the watershed part in Canada. The part in USA accounts for a quite large percentage of the total area (more than a quarter). It's unreasonable to ignore this part. (4) Line 128: "630 to 2400". The unit of the elevation should be added. (5) Section 2.2.1: Are the climatic data from the observed data or generated from climate models? (6) Lines 145: The multiplication sign cannot be represented by 'x'. (7) Section 3.1: This sub-section which just gives the data of forest disturbance should not be put in the method section. (8) Line 325: Why the parameter w in the Budyko-type equation is set to be 2? (9) Line 366: The detected breaking point in baseflow is in year 1972, but the most significant change in forest cover occurred in 1991 (see Line 204 in this paper). It seems that the change in baseflow has nothing to do with the forest disturbance. Why? (10) Figures: Why the number of figures in this paper starts from 11? It is inconsistent with the figure number in text. (11) Tables 1~4 cannot be found in this paper.

---

## Referee Comment (RC3) · Anonymous Referee #3 · 24 Jul 2016

This paper presented a study on the effect of forest disturbance and climate variability on baseflow changes in a forested watershed. The main finding is that both forest disturbance and climate variability have significant effects on baseflow magnitudes and patterns. The manuscript is on a topic of interest to the journal and the methodology may have practical values. However, the description of the methodology is confusing at parts and the logic in the result section needs to be improved. My suggestion would be major revision.

Specific comments:

1. P6, L11: The elevations here are missing units.

[Figure]

2. Figure 1: This figure only shows the part of watershed in Canada. Is the study also only considering the Canadian part of the watershed?

3. P11, L13-16: The definitions of Cbf and Cro are described twice here.

4. P15, L3-4: Please revise this part.

5. P15, L9: Did you mean "linear"?

6. P15, L14: I'm confused by the word "calibrated" here. Should it be "calculated"?

7. In general, the methodology part of the manuscript needs to be revised to improve the clarity. The order of the sections and how they link to each other may need to be better explained.

8. Figure 9: I assume the authors plot calculated groundwater discharge vs. calculated baseflow here to find breaking points that indicate baseflow changes. Even this method is described in previous studies, the authors may need to briefly explain the logic behind the method here. Also, what is "Pae"?

9. P19, L1-4: Based on figure 4 and 5, there is no significant forest disturbance in 1972. Is there any other major changes in that period?

10. The conclusion section need to be revised to provide a comprehensive summary of the study, in terms of methodology, discussion and general outcomes.

---

## Author Comment (AC1) · 11 Sep 2016

Reviewer's comments are in italic style.

*General Comments*

*This paper presented a study on the effect of forest disturbance and climate variability on baseflow changes in a forested watershed. The main finding is that both forest disturbance and climate variability have significant effects on baseflow magnitudes and patterns. The manuscript is on a topic of interest to the journal and the methodology may have practical values. However, the description of the methodology is confusing at parts and the logic in the result section needs to be improved. My suggestion would be major revision.*

**Response:** Thanks for the reviewer's efforts and comments on our manuscript. We agree that more detailed descriptions about the methods are needed. However, we must take a stance between too simple and too detailed descriptions as those methods have been applied and published by various studies. To address the reviewer's concern, we have added suitable details on the methods in the manuscript. Here are the revised descriptions of the methodology used in the manuscript.

Streamflow is divided into surface runoff ($RO$) and baseflow ($BF$). Theoretically, water balance in large watersheds can be expressed as:

$P = ET + RO + BF$ (3a) or $BF = P\text{-} ET\text{-} RO$ (3b).

In this study, effective precipitation ($P_e$) is then defined as the residual between precipitation after deduction of evapotranspiration and surface runoff. A linear relationship is assumed between accumulative baseflow ($BF_a$) and accumulative effective precipitation ($P_{ae}$) (Zheng et al., 2009; Wei and Zhang, 2010). Thus, the MDMC can be plotted by $BF_a$ against $P_{ae}$. In this way, the effects of climate variability on annual baseflow can be eliminated. For a period with no or little forest disturbance, a straight line is expected, which is acted as the baseline indicating the linear relationship between $BF_a$ and $P_{ae}$. Breaking points can be identified on the MDMC if there are significant influences from non-climatic variables, such as forest disturbance. The study period was

subsequently divided into reference and disturbance periods by the breakpoint. Once the statistical significance are found, the relationship between $BF_a$ and $P_{ae}$ in the reference period was therefore employed to predict accumulative annual baseflow for the disturbance period. The difference between the observed and predicted values was treated as the annual baseflow deviation caused by forest disturbance ($\Delta BF_f$). Thus, the annual baseflow deviation caused by climate variability can be determined as:

$$\Delta BF_c = \Delta BF - \Delta BF_f \tag{4}$$

Where, $\Delta BF$, $\Delta BF_f$ and $\Delta BF_c$ are the deviations of annual baseflow, annual baseflow deviation caused by forest disturbance and climate change, respectively.

**Specific comments:**

1.  *P6, L11: The elevations here are missing units.*

**Response:** We corrected the unit. The elevation ranges from 630 to 2400 meters above sea level.

2.  *Figure 1: This figure only shows the part of watershed in Canada. Is the study also only considering the Canadian part of the watershed?*

**Response:** Thank the reviewer for pointing this out. Our watershed is an international watershed that spans from Canada to USA with the flow in the US portion eventually draining into our hydrometric station in Canada. Unfortunately, data on forest disturbance from the US portion are not available. To consider this, we have checked available historic documents and data, and noticed that there are no major disturbance events occurred in the US part. The US part of watershed is located in the national parks, where forest logging is prohibited. Thus, we believe that our forest disturbance data on the Canadian part is

representative for the whole watershed. To address this concern, we have added some discussions on this uncertainty.

[Figure]

**Figure 1.** Location and elevations of the study watershed with the total area of 1810 km$^2$, of which 530 km$^2$ is in USA.

3. *P11, L13-16: The definitions of $C_{bf}$ and $C_{ro}$ are described twice here.*

**Response:** We have rephrased the sentences as follows.

$C_{ro}$ corresponds to the highest flow, while $C_{bf}$ corresponds to the lowest flow.

4. *P15, L3-4: Please revise this part.*

**Response:** We have revised this part in the method section. Please see our responses to the general comments for details.

5. *P15, L9: Did you mean "linear"?*

**Response:** Yes, this is a typo. We corrected it.

6. *P15, L14: I'm confused by the word "calibrated" here. Should it be "calculated"?*

**Response:** Yes, we changed the word "calibrated" to "calculated".

7. *In general, the methodology part of the manuscript needs to be revised to improve the clarity. The order of the sections and how they link to each other may need to be better explained.*

**Response:** Thanks for the suggestions. We have revised our descriptions on the methods (baseflow separation method and modified double mass curve). Please see our responses to the general comments for details.

8. *Figure 9: I assume the authors plot calculated groundwater discharge vs. calculated baseflow here to find breaking points that indicate baseflow changes. Even this method is described in previous studies, the authors may need to briefly explain the logic behind the method here. Also, what is "Pae"?*

**Response:** Thanks for pointing this out. We made a mistake in our calculations on determining the breakpoint of 1972 in the first version of our manuscript. We then re-calculated our data and determined that the new breaking point on MDMC was in the year of 1991, which coincides with forest disturbance history. The breaking point on MDMC were further confirmed by two breaking point tests (Table 3). Here are recalculated results.

[Figure]

**Figure 9.** Modified Double Mass Curve of cumulative annual baseflow vs. cumulative effective precipitation.

**Table 3.** Breaking point tests for the slopes of MDMC for baseflow

| Change Point | Pettitt test | | Z test | |
|---|---|---|---|---|
| | K | P | Z | P |
| Year 1991 | 389 | 0.032 | -3.39 | 0.001 |

**Table 4.** Relative contributions of forest disturbance and climate variability to annual baseflow in the Upper Similkameen River watershed from 1992 to 2013.

| Period | $\Delta BF$ | $\Delta BF_f$ | $\Delta BF_f/BF$ (%) | $\Delta BF_c$ | $\Delta BF_c/BF$ (%) | $R_f$ (%) | $R_c$ (%) | CECA (%) |
|---|---|---|---|---|---|---|---|---|
| 1992-2003 | -5.6 | 4.6 | 5.9 | -10.2 | -13.0 | 36.5 | 63.5 | 14.6 |
| 2004-2013 | 2.7 | 11.7 | 14.2 | -9.0 | -10.9 | 61.4 | 38.6 | 24.6 |
| 1992-2013 | -1.8 | 7.8 | 9.8 | -9.7 | -12.0 | 50.3 | 49.7 | 18.3 |

9. *P19, L1-4: Based on figure 4 and 5, there is no significant forest disturbance in 1972. Is there any other major changes in that period?*

**Response:** Please see comment 8 for details.

*10. The conclusion section need to be revised to provide a comprehensive summary of the study, in terms of methodology, discussion and general outcomes.*

**Response:** Here are the revision on conclusions.

We concluded that forest disturbance significantly increased annual baseflow of about 7.8 mm, while climate variability decreased 9.7 mm for the period of 1992 to 2013. The relative contributions of forest disturbance and climate variability were 40.3% and 59.7%, respectively for the study period. In addition, forest disturbance also altered seasonal baseflow patterns by increasing the spring baseflow and decreasing the summer baseflow. All those hydrological effects on baseflow have important implications for sustaining water supply and aquatic systems, which should be carefully managed.

---

## Author Comment (AC2) · 11 Sep 2016

Reviewer's comments are in italic style.

**General comments:**

*The main aim of the paper is to reveal or distinguish the individual contributions of forest disturbance and climate variability on baseflow in a forest catchment. However, the manuscript was poorly organized that so many uncertainties are in their methods and discussions. The authors adopted several distinct methods in their study just like the baseflow separation method which seems to be critical in their study, and the MDMC method for quantifying the relative contributions of forest disturbance and climate variability and even Budyko method has also been adopted in this study. However, the discussions for each method were not enough, which lead to many uncertainties and questions to the readers. For example, the baseflow separation method, CMB developed based on Pinder and Jones [1969], was firstly proposed to investigate the variation in the runoff composition in three small drainage basins which are relatively simple in flow path and components compared with larger catchments. As to the MDMC method, it needs a logistic flow chart to describe the whole processes, and more details should be provided on how you calculate $\Delta Q_f$ and $\Delta Q_c$. Another important issue is that how climate has changed in this catchment based on the observations and has climate changes impacted the forest itself about the coverage rate, beetle infestation? The figures and tables are in poor quality. Based on above considerations, especially the poor organization and insufficiency of discussions on various adopted methods, I have to recommend "major revisions", would like to encourage resubmission.*

**Response:** We thank the reviewer for helpful and important comments. The first comment is about the *lack of detailed descriptions and discussions on the methods used in the study*. We agree that more detailed descriptions about the methods are needed. However, we must take a stance between too simple and too detailed descriptions as those methods have been applied and published by various studies. To address the reviewer's concern, we have added suitable details on the methods

in the manuscript. Please also see our responses in comments 4, 15 and 16 for CMB method and comment 18 for MDMC method.

The second comment is about *how climate has changed in this catchment based on the observations and has climate changes impacted the forest itself about the coverage rate, beetle infestation?* The key objective for this study is to separate the relative contributions of climate and forest disturbance to baseflow. While we agree that how climate change might affect forests and beetle infestations is an important scientific question, it is beyond the scope of this study. The data on how climate has changed in the watershed is presented in Figure 2.

**Specific Comments:**

*P means page, and L means line*

1.  *P1L14: what is the different between baseflow and groundwater discharge?*

**Response:** Streamflow is composed of surface runoff and baseflow. Baseflow is mainly from the groundwater discharge to streamflow. In literature, baseflow and groundwater discharge are interchangeable concepts. In this manuscript, we have changed groundwater discharge to baseflow for consistency.

2.  *P1L17: how do you define a catchment a large one?*

**Response:** Various studies used different sizes to define large watersheds. The size of greater than 1000 $km^2$ is commonly used to define large watersheds, while the sizes of smaller than 1000 $km^2$ are for small watersheds. Our study took this definition, even though this is a subjective division.

3.  *P1L17-23: As you have mentioned "However, studying this topic is challenging as it requires explicit inclusion of climate into assessment due to their interactions at any large watersheds." How do you think that CMB method can solve this problem?*

**Response:** Studying the effects of forest disturbance and climate variability on baseflow is difficulty. The CMB method, in this study, was only used to separate the baseflow from streamflow, and was not used to separate the effects if forest disturbance and climate to hydrology. The baseflow in our study is mainly affected by both forest disturbance and climate variability. The modified double mass curve (MDMC) was then applied to separate the relative effects of forest disturbance and climate variability on baseflow.

4. *P2L11: 1) 'Introduction' is poorly written for that the method seems not to satisfy your main work and there so many uncertainties in your study. For instance, larger catchments tend to be disturbed by human activities in their wide downstream alluvial plain, and the hydrological responses seem to be lagged by drainage network and inapparent compared with small catchments. 2) As we know, CMB was developed based on scientific studies on smaller catchment, how do your upscale the method for applying in large scale catchment?*

**Response:** We agree that larger watersheds tend to be disturbed by human activities in their wide downstream alluvial plain, and the hydrological responses seem to be lagged by drainage network. This is why we studied the hydrological responses to forest disturbance and climate in a large watershed as it is likely different from those in small watersheds. Another advantage for selecting this large watershed is that forest change or disturbance is the only one dominant land cover change, while others (urban, agriculture etc.) are very minor and thus can be ignored. This advantage allows us easily separate the relative effects of climate and forest changes on hydrology (i.e. baseflow).

The CMB method was initially applied in small watersheds. The method was then applied in large watersheds (e.g. Miller et al., 2014 and 2015). Additionally, separating a hydrograph into two end-members (surface runoff and baseflow) is more appropriate for large watersheds, as compared with smaller ones where more than two sources should be considered (Uhlenbrook et al., 2002). Thus,

the CMB method was selected for this study. It is not our intention to upscale this method to large watersheds. To bring more clarity, we have rewritten this part of Introduction.

5. *P3L8: "...but had reached nearly 50% of the pre-cut level..." of WHAT?*

**Response:** Thanks. We corrected this as "… but groundwater table had reached nearly 50% of the pre-cut level"

6. *P4L4-7: "Although there are …annual streamflow…in large watersheds." What do you mean here? What is the logistic relation? I have expected as "There are many studies on annual streamflow but no study on monthly or daily streamflow". Anyway, critical comments in Introduction should be better if supported by data and comments of other colleagues.*

**Response:** Our intention is to say that the effects of forest disturbance and climate variability on annual streamflow have been extensively studied, but their effects on baseflow are rarely assessed. Here is our correction.

Although there are growing studies on separating the relative effects of forest change and climatic variability on hydrology, they are mainly on annual streamflow. As far as we know, there are no such studies to quantify the relative contributions of forest disturbance and climate variability to baseflow in large watersheds.

7. P4L10: "there are no commonly-accepted" As there are no commonly-accepted methods for baseflow separation, how do you verify that your adopted method is suitable for the study catchment.

**Response**: Numerous baseflow methods have been developed and verified (see manuscript for details). They can be lumped into two broad categories, i.e., tracer-based and non-tracer-based method. Compared to non-tracer methods, the tracer-based methods are considered as an objective

and physical-based method. Thus, the tracer-based methods are more preferable when tracer data are available. However, tracer-based methods are often limited by long-term data availability. In our study watershed, long-term data on streamflow and conductivity are available. Therefore, this method was selected for this study. We have rephrased our statement as follows.

The first challenge is the application of objective baseflow separation methods as they are often limited by long-term data requirement.

8. *P6L12: "...Manning Park drains to Okanogon River in U.S.A." I found that the river drains to Canada in figure 1?*

**Response**: Our study domain is the upper reach of the Smilkameen River. The lower reaches of Similkameen River drains to Okanogon River. We redraw the Figure 1.

9. *P6L16-18: "The bedrock types are generally resistant to water erosion, and form uplands and mountain ranges, which may contain bedrock aquifers, where are highly fractured." It seems to be inconsistent!! As it hard to be eroded, why is fractured and contains aquifers?*

**Response**: The fractures are not formed by water erosion, and they may be caused by compression and tensile from different bedrock layers (Sheets and Kozar, 2000).

10. *P8L1: how long have the forest disturbance databases been recorded?*

**Response**: The forest disturbances (i.e. logging, fire, and insect infestation) have been recorded from 1954 to 2009 in our study watershed. The equivalent clear-cut area (ECA) is calculated for this period. Please refer to Figures 4 and 5 in the manuscript for details.

**11. *P8L20: "…a total of 823…" Is the water quality data measured per month?***

**Response**: Before 1987, the measurements of conductivity were irregular. Two or three measurements were taken in some months, while only one or even none measurement were taken in others. The regular measurements, twice per month, have been conducted since 1988.

12. *P9L11: how about the distribution of forest disturbance? Is it changed every year? How to evaluation the change of position of forest disturbance on baseflow?*

**Response**: The forest disturbances were scattered throughout the study watershed. Yes, it is changed every year. Using our calculation of ECA (equivalent clear-cut area), we are able to calculate cumulative forest disturbance over time (see the manuscript for details). Regarding your question on *evaluation of the change of position of forest disturbance on baseflow,* this is beyond the scope of this study, but it could be an interesting question for future study.

13. *P11L8: CMB was developed based on scientific studies on smaller catchments. While applying in large catchments, other factors and processes, e.g., effects of exchange between riparian and river, storage and lag in river network, shall be considered. So CMB method in equation (1) may have to be improved before applying in large catchments.*

**Response**: Please see the response for comments or points 4, 15, and 16.

14. *P12L12-18: all the variables should be in italic.*

**Response**: Accepted.

15. *P13L12-16: this paragraph is difficult to be understood and need to be re-written.*

**Response**: We have rewritten the paragraph as follows:

The accuracy of baseflow estimation using the CMB method is highly dependent on the selection of two parameters ($C_{bf}$ and $C_{ro}$) (Equation 1). In this study, the annual paired $C_{bf}$ and $C_{ro}$ were selected for each water year (i.e. August to September) to minimize temporal variations in

conductivity rather than assuming the constant $C_{bf}$ and $C_{ro}$ through the whole study period (Zhang

et al., 2013; Li et al., 2014). Conductivities corresponding to 0.01 quantile flow rates (i.e. flow rates

exceed at 1% of the time in a given year) were averaged and treated as annual $C_{ro}$, while

conductivities corresponding to 0.99 quantile flow rates (i.e. flow rates are smaller than 99% of the

time in a given year) in one year were averaged and treated as annual $C_{bf}$, respectively.

16. *P13L19: As you only use one equation (Equation (1)) for baseflow separations, I could not*
    *figure out why will $C_{ro}$ play so different effects on baseflow in small and large catchments? As*
    *the smaller or larger value of $C_{ro}$ will impact the value of BF no matter what values of Q is*
    *adopted in terms of the scale of catchment areas. It is worth to note that $C_{ro}$ determines the*
    *ratio of BF to Q.*

**Response**: The original sentence is about *in-situ conductivity measurements in 14 large watersheds*

*(>1000 km$^2$) indicating that constant $C_{ro}$ might not have large impact on baseflow.* In Equation 1,

there is no doubt that either $C_{bf}$ or $C_{ro}$ can affect baseflow. However, using constant $C_{ro}$ for baseflow

separation might not have large impact on baseflow because: 1) the field measurement indicates

that temporal variation of $C_{ro}$ is smaller than $C_{bf}$ (e.g. Zhang et al., 2013; Millier et al., 2014 and

2015); and 2) the sensitivity of baseflow to $C_{ro}$ is smaller than $C_{bf}$ (Zhang et al., 2013; Li et al.,

2014).

17. *P14L15: Add a flow chart about MDMC.*

18. *P15L6-9: shall equation (3) be placed in individual line? How do you calculate Ge and BFa*
    *in equation (3)?*

**Response:** We added the more calculation descriptions for MDMC. The more detailed descriptions

for the method can also be found in our previous publications (e.g., Wei and Zhang et al., 2010;

Yao et al., 2012; Zhang et al., 2012; Zhang et al., 2014; Liu et al., 2015). For your convenience, we

briefly described the MDMC method below. In addition, we found that using effective groundwater discharge ($G_e$) might cause confusion for readers so we replaced it with effective precipitation ($P_e$). Streamflow is divided into surface runoff ($RO$) and baseflow ($BF$). Theoretically, water balance in large watersheds can be expressed as:

$P = ET + RO + BF$ (3a) or $BF = P- ET- RO$ (3b).

In this study, effective precipitation ($P_e$) is then defined as the residual between precipitation after deduction of evapotranspiration and surface runoff. A linear relationship is assumed between accumulative baseflow ($BF_a$) and accumulative effective precipitation ($P_{ae}$) (Zheng et al., 2009; Wei and Zhang, 2010). Thus, the MDMC can be plotted by $BF_a$ against $P_{ae}$. In this way, the effects of climate variability on annual baseflow can be eliminated. For a period with no or little forest disturbance, a straight line is expected, which is acted as the baseline indicating the linear relationship between $BF_a$ and $P_{ae}$. Breaking points can be identified on the MDMC if there are significant influences from non-climatic variables, such as forest disturbance. The study period was subsequently divided into reference and disturbance periods by the breakpoint. The relationship between $BF_a$ and $P_{ae}$ in the reference period was therefore employed to predict accumulative annual baseflow for the disturbance period. The difference between the observed and predicted values was treated as the annual baseflow deviation caused by forest disturbance ($\Delta BF_f$). Thus, the annual baseflow deviation caused by climate variability can be determined as:

$$\Delta BF_c = \Delta BF - \Delta BF_f \qquad (4)$$

Where, $\Delta BF$, $\Delta BF_f$ and $\Delta BF_c$ are the deviations of annual baseflow, annual baseflow deviation caused by forest disturbance and climate change, respectively.

19. *P16L5-14: what is the relationship between PET and MDMC?*

**Response**: In MDMC, the horizontal axis is the effective precipitation, which is the residual of precipitation after deduction of evapotranspiration and surface runoff. To calculate evapotranspiration, PET is needed as it is one of the critical parameters for determining evapotranspiration.

20. *P16L11: Budydo should be Budyko. w is ?*

**Response**: Corrected.

21. *P16L16: how do you determine the values of $C_{ro}$ and $C_{bf}$ in Section 4.1? As you defined them as constant, thus how do you derive their values?*

**Response**: $C_{ro}$ and $C_{bf}$ are not constant throughout the study period. In this study, the annual paired $C_{bf}$ and $C_{ro}$ were selected for each water year (i.e. August to September) to minimize temporal variations in conductivity rather than assuming the constant $C_{bf}$ and $C_{ro}$ through the whole study period (Zhang et al., 2013; Li et al., 2014). Conductivities corresponding to 0.01 quantile flow rates (i.e. flow rates exceed at 1% of the time in a given year) were averaged and treated as annual $C_{ro}$, while conductivities corresponding to 0.99 quantile flow rates (i.e. flow rates are smaller than 99% of the time in a given year) in one year were averaged and treated as annual $C_{bf}$, respectively.

*P18L14: in figure 9, I really can hardly figure out the breaking point at 1972. The authors need to display more data in figures or tables. What is the difference between the baseflow and groundwater discharge in the vertical and horizontal axis in figure 9?*

**Response**: Thanks for pointing this out. We made a mistake in our calculation on determining the breakpoint of 1972 in the first version of our manuscript. We then re-calculated our data and determined that the new breaking point on MDMC was in the year of 1991, which coincides with forest disturbance history. The breaking point on MDMC were further tested by two breaking point

tests (Table 3 and Comment No. 18). The difference between the baseflow and groundwater discharge in Figure 9 was addressed in Comment No. 18. Here are recalculated results.

[Figure]

**Figure 1.** Modified Double Mass Curve of cumulative annual baseflow vs. cumulative effective precipitation.

**Table 1.** Breaking point tests for the slopes of MDMC for baseflow

| Change Point | Pettitt test | | Z test | |
|---|---|---|---|---|
| | K | P | Z | P |
| Year 1991 | 389 | 0.032 | -3.39 | 0.001 |

**Table 2.** Relative contributions of forest disturbance and climate variability to annual baseflow in the Upper Similkameen River watershed from 1992 to 2013.

| Period | $\Delta BF$ | $\Delta BF_f$ | $\Delta BF_f/BF$ (%) | $\Delta BF_c$ | $\Delta BF_c/BF$ (%) | $R_f$ (%) | $R_c$ (%) | CECA (%) |
|---|---|---|---|---|---|---|---|---|
| 1992-2003 | -5.6 | 4.6 | 5.9 | -10.2 | -13.0 | 36.5 | 63.5 | 14.6 |
| 2004-2013 | 2.7 | 11.7 | 14.2 | -9.0 | -10.9 | 61.4 | 38.6 | 24.6 |
| 1992-2013 | -1.8 | 7.8 | 9.8 | -9.7 | -12.0 | 50.3 | 49.7 | 18.3 |

22. *P23L2-3: in table 3, cross-correlation results revealed that forest disturbance has altered the baseflow regimes, specifically, increased spring baseflow while decreased summer baseflow in our watershed? I also cannot find the trends.*

**Response**: This is the wrong citation of Table 3. The correct citation should be Table 2.

23. *P24L1: provide a figure to show how the climate changes or fluctuates.*

**Response**: Data on the long-term average precipitation and temperature is provided in Figure 2.

24. *P25L15: conclusions is too short and a deepen summary and perspective are needed. The references should be carefully revised according to requirements of HESS. Some of the references are lack of Doi (e.g., Weijs et al., 2013 and Winkler et al., 2014) and others may use wrong symbols (e.g., Power et al., 1999). Figure 1 should be revised by adding a complete map about the catchment though it belongs to two countries. Land cover map also needs to be added to show the forest distributions.*

**Response**: Thanks for the comments. We have rewritten the conclusions and revised the citations in the manuscript. The part of the watershed located in the USA was updated in the Figure 1. We will add forest harvesting and mountain pine beetle infestation map in our watershed as supplementary. The following are the revised conclusions.

We concluded that forest disturbance significantly increased annual baseflow of about 7.8 mm, while climate variability decreased 9.7 mm for the period of 1992 to 2013. The relative contributions of forest disturbance and climate variability were 40.3% and 59.7%, respectively for the study period. In addition, forest disturbance also altered seasonal baseflow patterns by increasing the spring baseflow and decreasing the summer baseflow. All those hydrological effects on baseflow have important implications for sustaining water supply and aquatic systems, which should be carefully managed.

25. *Figure 5, in this figure, I find that CECA in some curves have little decreases as a function of year. Please try to explain why this happens.*

**Response**: Yes, the CECA is an integrated parameter that describes the cumulative forest disturbance in a watershed. The CECA increases with the forest disturbance and decreases with the forest recovery. For some periods with no or little forest disturbance, the CECA decreases with the forest recovery.

26. *Figure 7, it is lack of BFI in the legend.*

**Response**: Corrected.

27. *Figure 9, what is the difference between baseflow and groundwater discharge?*

**Response**: We used the effective precipitation instead of the groundwater discharge.

28. *Figure 10, how do you compute $\Delta BF_f$? Why not add $\Delta BF_c$ in the same figure? Is $\Delta BF$ a bias between 1973 and 2013 and the reference period (before 1972)? There are not logging or beetle infestations in the reference period, are there? If no, how do you define it as the reference period?*

**Response**: We provided explanations about how to calculate $\Delta BF_f$. Please see comments No. 18 and 22 for references. We also added the figure for $\Delta BF_c$ for the period of 1992-2013.

[Figure]

**Figure 2**. Annual variations of the effects of forest disturbance on annual baseflow in the Upper Similkameen River watershed from 1992 to 2013.

[Figure]

**Figure 3**. Annual variations of the effects of climate variability on annual baseflow in the Upper Similkameen River watershed from 1992 to 2013.

References

Sheets, C.J. and Kozar, M.D.: Ground-water Quality in the Appalachian Plateaus, Kanawha River Basin, West Virginia. US Department of the Interior, US Geological Survey. 2000.

---

## Author Comment (AC3) · 11 Sep 2016

Reviewer's comments are in italic style.

**General comment**

*In this paper, the authors applied the Budyko based method to analyze the effect of climatic variability and forest disturbance on the annual runoff, and then separate the change of the baseflow from the total change of the annual runoff by using conductivity mass balance method. In general, it's hard to say that this paper has any strong innovation point, since it is just a case study based on two common methods. The authors just consider the influence of climatic variability and forest disturbance on the rainfall-runoff relationship, but ignore the possible variation in the relationship between baseflow and total runoff caused by the forest disturbance. The results of paper are questionable. Leaving aside the lack in the methodological contribution, this paper also has a very bad presentation. I think this paper cannot be accepted by this journal.*

**Response:**

We thank Reviewer 2 for the comments on our manuscript. We agree that our manuscript can be further improved with a better presentation and more clarifications on the methods. However, after carefully reviewing the comments from this reviewer, who recommended "rejection", we think that the recommendation is not justified. The followings are our responses.

Firstly, Reviewer 2 did not fully capture the research approach of our manuscript. The main objective of our manuscript is to distinguish the cumulative effects of forest disturbance (e.g. logging, fire, and insect infestation) and climate variability on baseflow. To accomplish this objective, the modified double mass curve (MDMC) was applied instead of using the Budyko based method. In our manuscript, the Budyko based method was only used to calculate the actual

evapotranspiration. As far as we know, it is impossible to apply the Budyko based method directly to quantify the effects of forest disturbance and climate change on either baseflow or streamflow. In short, we did not apply the Budyko based method for studying our key objective as stated in the reviewer's comment.

Secondly, the reviewer indicated that the manuscript may not have any strong innovation point on methods. We agree that our study did not develop any new methods which was not our key objective. Instead, we focused on the application side of two existing methods. More importantly, separating the effects of forest disturbance and climatic variability to hydrology are normally aimed at annual streamflow. As far as we know, our study is the first one to separate the hydrological effects of forest disturbance and climate on baseflow in a large watershed. In addition, we further modified MDMC for its application on the baseflow. Thus, our study made the first attempt to quantify the effects of forest disturbance on baseflow in a large watershed.

15  Thirdly, Reviewer 2 also indicated that the relationship between baseflow and total runoff were not considered. In fact, the relationship was fully considered in our study. The conductivity mass balance method (CMB) is an objective and physical-based baseflow method (see section 3.2 of the manuscript). To implement this method, the regression model (Equation 2) was used to estimate the long-term continuous daily conductivity data (see section 3.2.2). Then, estimated daily

20  conductivity data were adopted for baseflow separation. In Equation 2, the simple Fourier series and flow anomalies (i.e. 1 year, 30 days, and 1 day) were added to address the temporal variations of forest disturbance on conductivity data. Additionally, the paired parameters for the CMB method (i.e. conductivities of surface runoff and baseflow) were also selected for each year rather

than using constant values for the whole study period to overcome the temporal variations of conductivity. Thus, we believe that the relationship between baseflow and streamflow was fully considered in our study.

5      Finally, we feel that Reviewer 2 may not find the correct materials of the manuscript. For example, as indicated from comments 10 and 11, Reviewer 2 did not find any tables. In our manuscript, the captions of figures and tables are assorted in an ascendant order. All tables are listed at the end of the manuscript. Please see the original PDF file in the following link to download manuscript (http://www.hydrol-earth-syst-sci-discuss.net/hess-2016-291/hess-2016-291.pdf).    Additionally,
10     the reviewer questioned several terms (e.g., forest disturbance) and methods, which are widely used in ecohydrology (please see comments 1, 2, and 8).

**Specific comments:**

*(1) Line 31: "85.2_21.5", "0.22_0.05". I wonder how the terms 21.5 and 0.05 were calculated.*
15     *The authors presented neither the calculation method nor the specific meaning of these terms in the paper. The similar issues in other parts of this paper are also obvious.*

**Response:** The term of $85.2 \pm 21.5$ mm and $0.22 \pm 0.05$ are respective annual average baseflow and baseflow index (BFI), and their standard deviations for the period of 1954-2013. The calculations were done with Microsoft Excel.  They are the common statistics describing the
20     average values and deviations.

*(2) Line 35: What is the specific meaning of forest disturbance? Increase or decrease in forest cover?*

**Response:** Forest disturbance is the common term in ecology, and refers to the decrease in forest cover. Forest disturbances are the events (e.g., wildfire, harvesting, insect infestation) that cause the changes in forest growth, structures, and ecosystem functions.

5 *(3) Section 2: The study watershed in this paper spans USA and Canada, but the authors just consider the watershed part in Canada. The part in USA accounts for a quite large percentage of the total area (more than a quarter). It's unreasonable to ignore this part.*

**Response:** Thank the reviewer for pointing this out. Our watershed is an international watershed that spans from the Canada to the USA with the flow in the US portion eventually draining into

10 our hydrometric station in Canada. Unfortunately, data on forest disturbance from the US portion are not available. To consider this, we have checked available historic documents and data, and noticed that there are no major disturbance events occurred in the US part. The US part of watershed is located in the national parks, where forest logging is prohibited. Thus, we believe that our forest disturbance data on the Canadian part is representative for the whole watershed. To

15 address this concern, we have added some discussions on this uncertainty.

*(4) Line 128: "630 to 2400". The unit of the elevation should be added.*

**Response:** Elevation is described in meters above sea level.

20 *(5) Section 2.2.1: Are the climatic data from the observed data or generated from climate models?*

**Response:** Climate data were generated from the ClimateBC, which is a standalone program. It extracts and downscales PRISM monthly climate data to scale-free point locations. We used the resolution of $800 \times 800$ m climate data for our study. Please see Section 2.2.1 for details.

*(6) Lines 145: The multiplication sign cannot be represented by 'x'.*

**Response:** Accepted.

*(7) Section 3.1: This sub-section which just gives the data of forest disturbance should not be put in the method section.*

**Response:** Forest disturbance is calculated through original forest inventory data, which is not simple forest cover data. In this study, equivalent-clear-cut area (ECA) was adopted as proxy of forest disturbance. ECA is an integrated indicator that considers the forest disturbance and hydrological recovery at the different biogeoclimatic zones, elevations, species and etc. Thus, we placed the forest disturbance calculation in the *Methods* section. To address this comment, we have shorten our descriptions.

*(8) Line 325: Why the parameter w in the Budyko-type equation is set to be 2?*

**Response:** This equation (Equation 8) is developed by Zhang et al. (2001). According to Google Scholar, this publication has been cited more than 1250 times. This equation has been validated through 250 global catchments and widely used to calculate evapotranspiration at the catchment scale. The suggested plant-available water coefficient ($w$) for forest is 2 (Zhang et al., 2001). Given the large proportion of forest cover in our watershed, it is reasonable to set the $w$ to be 2.

*(9) Line 366: The detected breaking point in baseflow is in year 1972, but the most significant change in forest cover occurred in 1991 (see Line 204 in this paper). It seems that the change in baseflow has nothing to do with the forest disturbance. Why?*

**Response:** Thanks for pointing this out. We made a mistake in our calculations on determining the breakpoint of 1972 in the first version of our manuscript. We then re-calculated our data and determined

that the new breaking point on MDMC was in the year of 1991, which coincides with forest disturbance history. The breaking point on MDMC were further tested by two breaking point tests (Table 3). Here are recalculated results.

[Figure]

**Figure 9.** Modified Double Mass Curve of cumulative annual baseflow vs. cumulative effective precipitation.

**Table 3.** Breaking point tests for the slopes of MDMC for baseflow

| Change Point | Pettitt test | | Z test | |
| --- | --- | --- | --- | --- |
| | K | P | Z | P |
| Year 1991 | 389 | 0.032 | -3.39 | 0.001 |

**Table 4.** Relative contributions of forest disturbance and climate variability to annual baseflow in the Upper Similkameen River watershed from 1992 to 2013.

| Period | $\Delta BF$ | $\Delta BF_f$ | $\Delta BF_f/BF$ (%) | $\Delta BF_c$ | $\Delta BF_c/BF$ (%) | $R_f$ (%) | $R_c$ (%) | CECA (%) |
| --- | --- | --- | --- | --- | --- | --- | --- | --- |
| 1992-2003 | -5.6 | 4.6 | 5.9 | -10.2 | -13.0 | 36.5 | 63.5 | 14.6 |
| 2004-2013 | 2.7 | 11.7 | 14.2 | -9.0 | -10.9 | 61.4 | 38.6 | 24.6 |
| 1992-2013 | -1.8 | 7.8 | 9.8 | -9.7 | -12.0 | 50.3 | 49.7 | 18.3 |

*(10) Figures: Why the number of figures in this paper starts from 11? It is inconsistent with the figure number in text.*

*(11) Tables 1_4 cannot be found in this paper.*

**Response:** The captions of figures in this paper are in an assorted order. Please refer to the manuscript for details. Tables are listed at the end of the manuscript. Please see the original PDF files in the following link for details (http://www.hydrol-earth-syst-sci-discuss.net/hess-2016-291/hess-2016-291.pdf).

**References**

Wei, X., and M. Zhang (2010), Quantifying streamflow change caused by forest disturbance at a large spatial scale: A single watershed study, Water Resour. Res., 46, W12525, doi: 10.1029/2010WR009250.

Zhang, L., W. R. Dawes, and G. R. Walker (2001), Response of mean annual evapotranspiration to vegetation changes at catchment scale, Water Resour. Res., 37(3), 701–708, doi: 10.1029/2000WR900325.